# The Role of Neurorehabilitation in Post-COVID-19 Syndrome

Lara Diem * , Oliver Höfle, Livia Fregolente and Robert Hoepner

Department of Neurology, Inselspital, Bern University Hospital, University of Bern, Freiburgstrasse, 3010 Bern, Switzerland

* Correspondence: larafrancesca.diem@insel.ch

**Abstract:** Post-COVID-19 syndrome is an ongoing challenge for healthcare systems as well as for society. The clinical picture of post-COVID-19 syndrome is heterogeneous, including fatigue, sleep disturbances, pain, hair loss, and gastrointestinal symptoms such as chronic diarrhea. Neurological complaints such as fatigue, cognitive impairment, and sleep disturbances are common. Due to the short timeframe of experience and small amount of evidence in this field, the treatment of post-COVID-19 syndrome remains a challenge. Currently, therapeutic options for post-COVID-19 syndrome are limited to non-pharmaceutical interventions and the symptomatic therapy of respective symptoms. In this article, we summarize the current knowledge about therapeutic options for the treatment of neurological symptoms of post-COVID-19 syndrome.

**Keywords:** SARS-CoV-2; coronavirus; post-infectious; long-term symptoms; neuropsychiatric symptoms; viral infection; therapy; rehabilitation

## 1. Introduction

Severe acute respiratory syndrome coronavirus type 2 (SARS-CoV-2), which emerged in 2019, still presents medical, societal, and economic challenges.

In the beginning, the main medical considerations were the acute effects and consequences of this infection; later, however, the extent of the long-term consequences also became apparent. Indeed, not every patient fully recovers from the 2019 coronavirus disease (COVID-19) within weeks or months, resulting in post-COVID-19 syndrome. The World Health Organization (WHO) has established an official definition for post-COVID-19 syndrome, i.e., the persistence of symptoms that cannot be explained by another disease 3 months after infection [1].

The pathogenesis of post-COVID-19 syndrome is still unclear. Current evidence suggests that several mechanisms may be involved in the pathogenesis of post-COVID-19 syndrome: (1) Virus-related cellular changes associated with neurotropism. This mechanism could be responsible for olfactory dysfunction or autonomic nervous system dysfunction. (2) A dysregulated immune response in response to the initial infection, including autoimmune manifestations, activation of coagulation and fibrosis pathways, or metabolic disturbances. An increase in multiple inflammatory mediators has also been demonstrated in patients with post-COVID-19 syndrome. (3) Occult viral persistence [2].

According to recent data from the UK National Institute of Health, 3% to 11.7% of the infected suffer from post-COVID-19 syndrome [3]. A recent study by the Global Burden of Disease Long COVID Collaborators estimated that 6.2% of the patients with symptomatic SARS-CoV-2 infections who survived the acute phase in 2020 and 2021 have experienced post-COVID-19 syndrome.

The clinical picture of post-COVID-19 syndrome is heterogeneous, including fatigue, sleep disturbances, pain, hair loss, and gastrointestinal symptoms such as chronic diarrhea [4–9]. The most frequent symptom of post-COVID-19 syndrome is fatigue. Recent studies have shown a fatigue frequency of 50–90% in patients with post-COVID-19 syndrome [5,7].

The evidence for pharmacological and causal therapy in post-COVID-19 syndrome is still limited. However, several drug trials for the treatment of post-COVID-19 syndromes are ongoing, including, for example, temelimab (NCT05497089), montelukast (NCT04695704), lithium (NCT05618587), nirmatrelvir plus ritonavir (NCT05576662), low-dose naltrexone (NCT05430152), and fampridine (NCT05274477), among others. Currently, therapeutic options for post-COVID-19 syndrome are limited to non-pharmaceutical interventions such as physiotherapy (pacing and heart rate monitoring), occupational therapy (energy management education), and the symptomatic therapy of respective symptoms such as the treatment of headaches and neuropathic pain.

In this article, we summarize the current knowledge of therapeutic options for treating the neurological symptoms of post-COVID-19 syndrome.

Furthermore, we would like to emphasize that the recommendations regarding post-COVID-19 syndrome are constantly changing; thus, it is strongly recommended that the reader consider this work a summary of the current knowledge at the time of publication and keep up to date with the latest national and international recommendations. We expect Swiss-specific guidelines to be published by the Federal Office of Public Health sometime May 2023.

## 2. Neurological Manifestations of Post-COVID-19 Syndrome

### 2.1. Fatigue

Fatigue, as is known from studies of other post-infectious syndromes, directly affects quality of life and participation in the activities of daily living (ADL) [8,10]. Fatigue is defined as a debilitating feeling of mental and/or physical loss of energy [11] and can be accompanied, especially in post-COVID-19 syndrome, by a post-exertional malaise (PEM) [12]. PEM refers to the worsening of symptoms after physical, mental, or emotional exertion; it is also referred to as a "crash" [13]. This is present in more than 50% of patients with fatigue [14], and in the view of the authors, differentiates the fatigue associated with post-COVID-19 syndrome from fatigue of other origins, such as multiple sclerosis-related fatigue [15].

### 2.2. Cognitive Symptoms

In a survey study by Davis et al., 85.1% of respondents reported cognitive dysfunction, including poor attention and executive functioning (such as problem solving and decision making) [8].

Cognitive deficits, which are frequently found in patients with post-COVID-19 syndrome, relate to planning, thinking, concentration, and memory and/or language performance. In a study by Ortelli et al., patients with post-COVID-19 syndrome showed deficits in the Montreal Cognitive Assessment (MoCA) and Frontal Assessment Battery (FAB) for global cognition and executive functions, as well as in computerized attentive tasks for evaluating vigilance and executive attention compared to healthy individuals (age-and sex-matched healthy volunteers without COVID-19) [11]. Current test recommendations for short cognitive tests in clinical routines include the Symbol Digit Modalities Test (SDMT; for procession speed) or the Demenz Detection Test (DemTEC; according to the recommendation of the Swiss Society of Insurance Medicine) [16,17]. The MoCA is likely to be poor at identifying cognitive deficits in patients with post-COVID-19 syndrome [18]. However, it must be kept in mind that these tests are not validated for post-COVID-19 syndrome and should only be used as an aid alongside history and clinical assessment for the detection of cognitive deficits. Otherwise, detailed neuropsychological testing is recommended [16].

Patients' quality of life and ability to work are influenced by symptoms such as fatigue and cognitive deficits. The data on the incapacity to work due to post-COVID-19 syndrome also vary widely depending on the literature, and the percentage of people experiencing an incapacity to work ranges from 23.3 to 62% [8,9,19]. This difference is most likely based on the definition of incapacity (partial vs. total). In the study by Davis et al., 23.3% of patients were totally unable to work and 48% were partially unable to work, whereas in our study,

the percentage of people partially unable to work was 62% [8,19]. In general, the inability to work usually lasts longer than 13 weeks [8,9,19].

### 2.3. Other Frequent Symptoms

Other frequently reported post-COVID-19 symptoms are headache; pain, such as arthralgia; muscle pain; neuropathic pain; and sleep disturbances [10,19–21]. Neuropathic pain was also previously reported in patients who recovered from SARS-CoV-1, a virus belonging to the same family of coronaviruses, which caused an outbreak in 2003 [21].

Concerning sleep disturbances, quite similar results are found in the literature, with a prevalence of sleep disturbances of more than 50% [19,22]. The mean Insomnia Severity Index (ISI) was 13.0 points, and more than 1/3 of the respondents were above the threshold for clinical insomnia (defined by the ISI as ≥15 points) [19,22]. Excessive daytime sleepiness (defined by the Epworth Sleepiness Scale (ESS) as ≥11 points) was reported in 56.3% of the respondents [19]. Given such elevated ESS scores, obstructive sleep apnea syndrome (OSA) must be considered in the differential diagnostic workup for post-COVID-19 syndrome. In fact, a current study has shown a high rate of OSA in COVID-19-patients that exceeds the expected prevalence of OSA [23].

Autonomic dysfunction, e.g., postural tachycardia syndrome (POTS), also occurs in post-COVID-19 syndrome. Depending on the literature, the frequency of tachycardia syndromes, including POTS, varies between 9 and 50% [10,23–25]. The assessment of these autonomic symptoms, e.g., with the Schellong test, is particularly important, as symptoms such as postural tachycardia syndrome can be very limiting and can exacerbate fatigue.

## 3. Treatment of Post-COVID-19 Neurological Symptoms

Currently, therapeutic options for post-COVID-19 syndrome are limited to non-drug measures or the symptomatic therapy of symptoms. One of the greatest therapeutic challenges is treating neurological symptoms.

In this section, we focus on the treatment of the three most frequent neurological symptoms: fatigue, pain, and sleep disorders.

In the last 2 years, there has been an increase in the number of studies on non-drug interventions for fatigue [26,27]. In addition, lessons could be learned from the experience of treating fatigue in other conditions such as multiple sclerosis, chronic fatigue syndrome, and cancer-associated fatigue. However, we have to be very careful, because not all known therapeutic options for fatigue can be applied to post-COVID-19 fatigue. In contrast to the role of exercise in MS fatigue, in post-COVID-19 patients, due to post-exertional malaise, the anaerobic threshold does not need to be exceeded.

For the treatment of the other symptoms, we also refer to the recommendations of other guidelines from different countries, such as the NICE guidelines (UK) or S1 guidelines (Germany) [28,29].

Neurorehabilitation can take place in an outpatient or inpatient setting. Initially, outpatient options should be prioritized. Inpatient rehabilitation should be evaluated if there is no improvement in symptoms following outpatient therapy. Inpatient rehabilitation can offer a more intensive and multimodal approach to therapy. Due to the variability of post-COVID-19 syndrome, a multimodal therapy approach such as inpatient rehabilitation can be advantageous.

### 3.1. Occupational Therapy

Occupational therapy is a cornerstone of post-COVID-19 syndrome rehabilitation. In MS-associated fatigue, occupational therapy with an inpatient energy management education (IEME) program showed good effects [30–32]. IEME integrates the principles of patient education, the trans-theoretical model of behavior change, social cognitive theory, energy conservation strategies, and cognitive behavioral technique (CBT) [30,31]. This therapy is targeted at helping patients to better manage their symptoms and the negative impact such symptoms have on activities of daily living (fatigue, pain, cognitive

impairment, etc.). Furthermore, the patients should use the 3Ps in everyday life: planning, prioritization, and pause management. Finally, the main goal is to maintain autonomy and increase quality of life despite ongoing post-COVID-19 symptoms. The outpatient and inpatient versions of energy management education (EME) consist of the same eight topics and use the same behavior change techniques. They differ in the frequency of lessons and self-training tasks [33] (Figure 1). It should be noted that EME has been successfully used for MS fatigue but has not yet been validated in post-COVID-19 syndrome. However, it should be used in the clinical routine of post-COVID-19 patients.

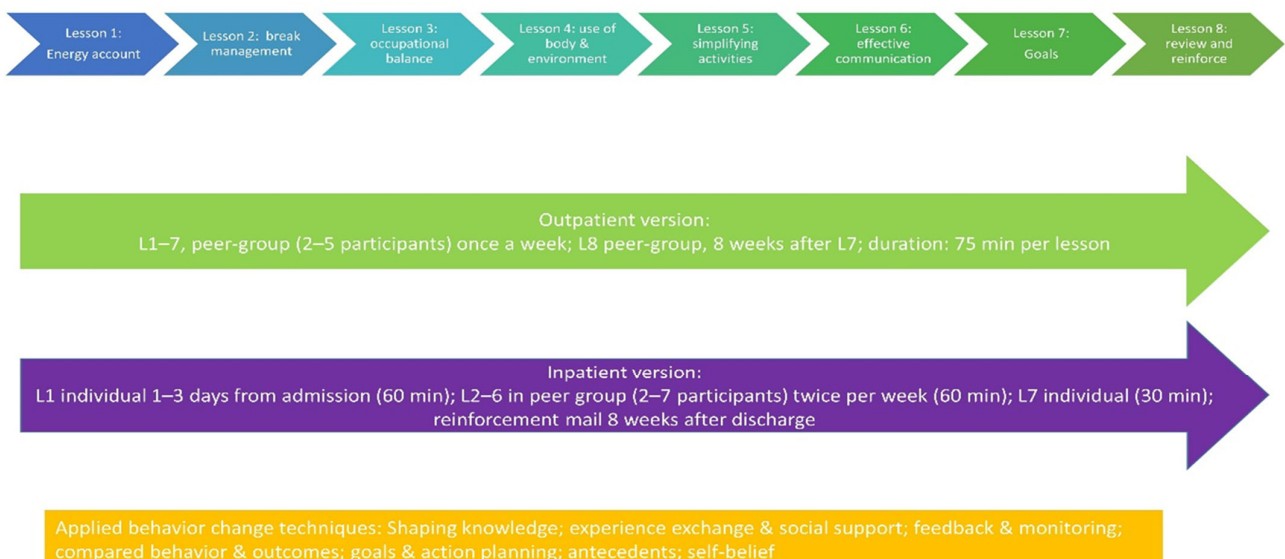

**Figure 1.** Description of energy management education (EME) [30].

CBT for severe fatigue has been found to be effective in several randomized controlled trials [34–36]. A recent multicenter two-arm randomized controlled trial (ReCOVer study) tested the efficacy of a program with CBT in patients experiencing severe post-infectious fatigue after a COVID-19 infection [37]. This program, named Fit after COVID-19, consists of up to nine modules ((1) goal setting; (2) sleep–wake pattern; (3) helpful thinking; (4) social support; (5) graded activity; (6) processing the acute phase of COVID-19; (7) fear and worries regarding COVID-19; (8) coping with pain; and (9) realizing goals). The results have not yet been published, but recruitment is completed and will represent a milestone in post-COVID-19 treatment. To stay up to date, we suggest using the International Clinical Trial Registry Platform (ICTRP).

### 3.2. Physiotherapy

#### 3.2.1. Graduated Exercise Therapy

A systematic review of graduated exercise therapy (GET) for chronic fatigue syndrome (CFS) concluded that patients with CFS generally feel less fatigued and their sleep and physical function improve after completing graduated exercise therapy [38]. Since the update of the NICE guidelines in 2020, GET is no longer recommended. The NICE guidelines do not recommend GET because this treatment is considered to be ineffective and harmful based on evidence from patient surveys and qualitative studies [39]. This decision has been widely criticized by the scientific community due to the choice of analyzed studies [40,41]. In addition, the omission of a recently published meta-analysis of the safety results from ten published studies on GET was strongly objected to [42]. This study showed that GET is safe as long as it is prescribed properly [42].

However, it is essential that physical activity be adjusted to the individual's limit to avoid triggering severe post-exertional malaise. It is therefore imperative to determine the limit of each patient before starting physiotherapy treatment. In the presence

of post-exertional malaise, activity management or pacing [43,44], as well as heart rate monitoring [45], may be effective rehabilitation approaches to support the self-management of symptoms.

### 3.2.2. Pacing

Pacing is a prudent use of resources on a physical, mental, and emotional level. The goal of pacing is to find the optimal and individual balance between rest periods and activation periods (physical, cognitive, and emotional) [43,44].

The sustained stabilization of symptoms, which often fluctuate, could guide how activities and rest are modified in response to the individual and frequently subjective symptoms. Quality of rest, sleep, and eating habits can also be considered as part of activity management and may help in the stabilization of symptoms.

Pacing should include setting realistic goals, monitoring physical, cognitive and social activities and their impact on energy levels, and avoiding possible overexertion that could exacerbate symptoms. Pacing is not an activity avoidance strategy, but a strategy to minimize the worsening of symptoms after exertion [44,46]. Pacing is often used as part of a set of energy conservation strategies called the "Principle of Three P's", namely, prioritization, planning, and pause management [47].

### 3.3. Heart Rate Monitoring

Heart rate monitoring is a rehabilitation strategy that can be used by people to self-manage symptoms when living with post-COVID-19 syndrome. Heart rate monitoring is a useful method for people with fatigue to avoid PEM. It refers to the continuous measurement of heart rate with a heart rate monitor, and it is used to more accurately pace daily activities and monitor the body's responses to exertion. The aim of heart rate monitoring is to stay below the ventilatory anaerobic threshold (VAT) throughout the day, thus avoiding post-exertional malaise [45]. The exact determination of VAT is carried out with the help of a cardiopulmonary exercise test (spiroergometry). However, this test can lead to post-exertional malaise in some patients. The pulse–threshold rate can also be calculated as an alternative. The Workwell Foundation recommends two methods for calculation [47]:

- $(220 - \text{age}) \times 0.55$ = guideline (in beats per minute (bpm));
- Resting heart rate as a measure for activity management. Resting heart rate can be determined by having the person lie flat in bed and calculating the average resting heart rate over 7 days. The benchmark is now set as 15 bpm above the resting heart rate.

Before using heart rate monitoring, autonomic dysfunction (e.g., postural tachycardia syndrome) in the context of post-COVID-19 syndrome must be ruled out, as this makes the use of these methods impractical.

### 3.4. Treatment of Neuropathic Pain

Non-medicinal measures including psychological support, exercise intervention, and patient education play an important role. As a general rule, if symptoms are frequent and other underlying causes—such as vitamin deficiencies, diabetes mellitus, iron deficiency or thyroid dysfunction—have been ruled out, neuropathies can be treated with pharmacological treatments including, but not limited to, pregabalin, gabapentin, or duloxetine, as well as local capsaicin treatment [48,49].

### 3.5. Treatment of Headaches

In the treatment of headaches, non-pharmacological measures such as progressive muscle relaxation, biofeedback, and patient education also play an important role. In particular, progressive muscle relaxation as defined by Jacobsen has shown positive effects on headaches, e.g., migraines. During this technique, a total of 16 muscle groups of the body are identified and slightly tensed for a few seconds, which is followed by a 30 to 40 s

relaxation phase. The practitioner concentrates on the perceived differences between the tension and the relaxation. In this way, the entire body is successively relaxed [50].

If there are no red flags [51], then symptomatic treatments of the acute headache attacks are suggested (such as non-steroidal anti-inflammatory drugs (NSAIDs), paracetamol, or triptans in post-COVID-19 headaches that meet migraine criteria; this is according to recommendations of the International Headache Society) [52–54]. However, it is very important to make the patient aware that painkillers should be taken for no more than 10–12 days per month in order to minimize the risk of an analgesic-induced headache [54].

In the event of chronic debilitating headaches which do not improve with standard analgesics, a disease-modifying treatment specific to the headache type should be instituted [53,54].

### 3.6. Treatment of Sleep Disorders

Although pharmacological therapy is commonly prescribed for insomnia, the current guidelines do not recommend it to be a first-line therapy, especially due to the adverse effects and a lack of lasting effects [55]. Current evidence suggests that cognitive behavioral therapy is an effective and cost–benefit-favorable alternative and should therefore be offered to patients suffering from insomnia [56]. The first steps that could already be implemented by the primary health provider would be reviewing sleep hygiene measures [55]:

(1) Do not drink caffeinated beverages (coffee, black tea, or cola) after lunch;
(2) Avoid alcohol to a large extent and do not use it as a sleeping pill under any circumstances;
(3) No heavy meals in the evening;
(4) Regular physical activity;
(5) Gradually reduce mental and physical exertion before going to bed;
(6) Introduce a personal bedtime ritual;
(7) Create a pleasant atmosphere in the bedroom (quiet and darkened);
(8) Do not look at an alarm clock or wristwatch during the night.

If pharmacologic treatment is needed for insomnia, phytotherapy (valerian, Avena sativa, and Passiflora, for example), melatonin, or sleep-inducing antidepressants (for example, mirtazapine or trazodone) may be prescribed for post-COVID-19 insomnia in addition to sleep hygiene measures over a short period. Benzodiazepines should be avoided in the treatment of sleep disorders [55].

**Author Contributions:** Conceptualization, L.D.; methodology, L.D.; validation, L.D., L.F., O.H. and R.H.; writing—original draft preparation, L.D.; writing—review and editing, L.D., L.F., O.H. and R.H. All authors have read and agreed to the published version of the manuscript.

**Funding:** This research received no external funding.

**Institutional Review Board Statement:** Not applicable.

**Informed Consent Statement:** Not applicable.

**Data Availability Statement:** No new data were created or analyzed in this study. Data sharing is not applicable to this article.

**Conflicts of Interest:** The authors declare no conflict of interest.

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
