# Peer review of "The Role of Neurorehabilitation in Post-COVID-19 Syndrome"

_ctn, doi:10.3390/ctn7020013_

Round 1

Reviewer 1 Report

Dear Dr Diem,

Thank you for giving me the opportunity to review ‘The role of neurorehabilitation in Post-COVID-19 syndrome’, a brief narrative description of the most common neurological symptoms following COVID-19 and management strategies for them.

The bar to acceptance in, by necessity, high, and requires unique work to provide new and novel findings. At the moment, unfortunately, I don’t believe this manuscript offers that. I appreciate the hard work that has gone into it, and can see some potential, but in its current form, I don’t believe it is ready for publication.

I have, therefore, suggested Major Revision to the Editor, and have provided detailed feedback as to the areas I would concentrate on.

Sentences are long, so would recommend restructuring or using more commas to break them up. Perhaps asking a native English speaker to review the manuscript might help this.

I am not sure that Post COVID-19 syndrome requires a capitalised P.

There are repeated mentions of your own studies, which, at times, make this appear like a rehash of your previous published work. You need to build on what you have already published, not just repeat it. Throughout the manuscript, you state facts without building on them – what is the relevance and implications for the work (both your work and other people’s) that you have cited? Some areas to think about should include some mechanisms and underlying reasons for the symptoms and deficits. If we understand underlying mechanisms, we are better able to design appropriate interventions, rather than just the treating the symptoms. You also need to discuss how these symptoms/limitations need to be assessed for? 

Good luck with the next stage,

Best wishes

Abstract – no comments

Introduction

Line 21: in parenthesis, the word ‘virus’ is redundant, as SARS-CoV-2 contains the word coronavirus 

Line 22: I would rephrase the end of this sentence, so it reads either, ‘still represents a multifactorial challenge’ or ‘still represents a medical, societal, and economic challenge’.

Line 26: please add 2019 to Coronavirus Disease. I would recommend reviewing this sentence, 25-28, as it is very long and the tense changes mid-way.

Line 45: You state one finding and use four references. Can you review this sentence please?

Line 47-48: I would review this sentence. It is not as true as it was two years ago. There is an incredible *volume* of evidence, some of which is very high quality (national/international RCTs, meta-analyses, etc), but overall, its reliability/validity/reproducibility is variable, and the longitudinal nature of it is not certain. National bodies, such as NICE in the UK, have made evidence-based recommendations on therapy.

Line 48-51: Likewise, as an extension of above, there are multiple pharmacological therapies undergoing validation and in trials, so I would rephrase to reflect that. It is not the same as two years ago. This revised message would still flow into the following sentence (52-53).

Line 58: there appears to be a rogue ‘2’ at the end of the sentence, I suspect this is a reference that has gone wrong.

Neurological manifestations of post-COVID-19 syndrome

Fatigue

Line 66: you can remove the ‘the’ preceding post-COVID-19 syndrome.

Line 72: I am not sure what this sentence adds

Cognitive symptoms

You describe fatigue a few times in this sentence, which is slightly repetitive to the section above.

Line 76: I would add ‘such’ to ‘as’ in parenthesis.

Line 80-82: What do these assessment tests and what is the conclusion gained from them?

Line 82-85: Why not MoCA and why SDMT and DemTec. You need to provide an explanation for these statements.

Line 86-87: Could you rephrase this sentence please, it is a bit clumsy.

Line 89: What was the reason for inability to work?

Could you describe some potential causes for the cognitive symptoms? Is it neural damage, specific impact of SARS-CoV-2 in the brain, the impact of fatigue, etc?

Other frequent symptoms

Line 93-96: some commas would help here to break up the sentence.

Line 95-100: These three sentences could be combined into one

Line 99: You have previously used the term SARS-CoV-2, so I recommend using SARS-CoV-1 for consistency.

Line 108-109: I am not sure what this adds – 12-40% is within the range of 9-50% previously mentioned.

Line 109-112: Which authors do you refer to? You? Also, you’ve already defined POTS so can use it. Again, overall, this is a long sentence that could be reduced to ‘As dysautonomia can exacerbate symptoms, including fatigue, it should be assessed for’. But you need to mention how. What is the best form of assessment?

Treatment of Post-COVID-19 neurological symptoms

Line 115: I don’t like the first sentence of the paragraph and suggest you remove it.

Line 115-123: Infact, I don’t like the entire paragraph. What are you trying to say? There isn’t a key message in this paragraph.

Line 119-120: I don’t think this is true. From where I am sitting, it appears there are multiple funders and many opportunities. Can you demonstrate or provide a citation please?

Line 127-139: This should be combined and act as your opening paragraph

Line 127: ‘the authors as board certified neurologists’ is an irrelevance.

Line 132: What are the implications regarding outpatient or inpatient settings? These are fundamentally different models with different levels of intensity, hours of rehabilitation, members of the multi-disciplinary team, resources…could you provide some discussion on one vs other?

Line 133: I disagree regarding rarity. There has been a systematic review and meta analysis on impact of rehabilitation on fatigue (de Sire, A.; Moggio, L.; Marotta, N.; Agostini, F.; Tasselli, A.; Drago Ferrante, V.; Curci, C.; Calafiore, D.; Ferraro, F.; Bernetti, A.; Ozyemisci Taskiran, O.; Ammendolia, A. Impact of Rehabilitation on Fatigue in Post-COVID-19 Patients: A Systematic Review and Meta-Analysis. Appl. Sci. 2022, 12, 8593. https://doi.org/10.3390/app12178593) and one performed on fatigue and cognitive impairment (Felicia Ceban, Susan Ling, Leanna M.W. Lui, Yena Lee, Hartej Gill, Kayla M. Teopiz, Nelson B. Rodrigues, Mehala Subramaniapillai, Joshua D. Di Vincenzo, Bing Cao, Kangguang Lin, Rodrigo B. Mansur, Roger C. Ho, Joshua D. Rosenblat, Kamilla W. Miskowiak, Maj Vinberg, Vladimir Maletic, Roger S. McIntyre, Fatigue and cognitive impairment in Post-COVID-19 Syndrome: A systematic review and meta-analysis, Brain, Behavior, and Immunity,

Volume 101, 2022, Pages 93-135, ISSN 0889-1591, https://doi.org/10.1016/j.bbi.2021.12.020.)

Occupational therapy

Line 141: In, not Im

Line 145-149: Check your tense

Line 150: I would specify that IEME is used successfully in MS-fatigue and could be used for post-COVID-19 fatigue. The way this is phrased suggests it has been designed for post-COVID-19 fatigue.

Line 157: Earlier you said there was no evidence based treatment for post-COVID-19, but here you cite an RCT

Line 161-162: You can delete this sentence

Physiotherapy 

GET

This is a highly debated area. The most recent NICE guidance for ME/CFS have removed GET. There are multiple systematic reviews with different outcomes. I recommend you look into this area further. Furthermore, you need to conduct a baseline assessment prior to commencement, which you haven’t noted. 

Pacing

Line 178: I don’t think that pacing involves increasing activity levels.

Line 188: This sentence should sit with 179-181

Heart rate monitoring

You have provided no detail as how to perform heart rate monitoring, nor how should it be used in combination with pacing or GET. This section needs to be developed to provide a practical discussion of how HR monitoring should be used in post-COVID-19 syndrome, not just stating that it has been used in ME/CFS. Where does heart rate variability fit in?

Neuropathic pain

Should pharmacological therapy sit alongside non-pharmacological therapy, including psychological support, exercise intervention, patient education, and an understanding how to live well with persistent or chronic pain?

Headache

What red flags? What medication regime/duration? What other elements need to be considered (such as PPI in chronic NSAID usage?) What do patients do on days when they can’t take medication? What other options are there?

Who or what is Jacobsen? Can you expand this area and provide some citations please?

Sleep

Please don’t start the paragraph on sleep with medication. That is the last option that should be considered. Start with the first steps discussing sleep hygiene, then CBT, etc, then medication. Please specify which medication, how they work, and what regime.

References

 All COVID-19 related references should be within the last 12 months given the volume of ongoing research and updated findings that have been published.

Reference 16 needs to be amended

Reference 22 is not a reference

Reference 23 needs to be amended

Reference 43 needs to be amended

Author Response

Response to reviewer 1

Dear Reviewer

Thank you for your valuable comments, which we appreciate. We hope we could address all your concerns properly.

  1. Line 21: in parenthesis, the word ‘virus’ is redundant, as SARS-CoV-2 contains the word coronavirus

Thank you for the comment. I have corrected the sentence accordingly.

Page 1, lines 21-22

The severe acute respiratory syndrome coronavirus type 2 (SARS-CoV-2), which emerged in 2019, still represents a medical, societal, and economic challenge’

  1. Line 22: I would rephrase the end of this sentence, so it reads either, ‘still represents a multifactorial challenge’ or ‘still represents a medical, societal, and economic challenge’.

See above

  1. Line 26: please add 2019 to Coronavirus Disease. I would recommend reviewing this sentence, 25-28, as it is very long and the tense changes mid-way.

Thank you for the comment. I have corrected the sentence accordingly.

Page 1, lines 24-25

Indeed, not every patient fully recovers from Coronavirus disease 2019 (COVID-19) within weeks to months leading to post-COVID-19 syndrome.

  1. Line 45: You state one finding and use four references. Can you review this sentence please?

Thank you for the comment. I have corrected the sentence accordingly.

Page 2, lines 51-52

Recent works have shown a fatigue frequency of 50%-90% in patients with post-COVID-19 syndrome. [5,7].

  1. Line 47-48: I would review this sentence. It is not as true as it was two years ago. There is an incredible *volume* of evidence, some of which is very high quality (national/international RCTs, meta-analyses, etc), but overall, its reliability/validity/reproducibility is variable, and the longitudinal nature of it is not certain. National bodies, such as NICE in the UK, have made evidence-based recommendations on therapy.

Thank you for the comment. While more data could be obtained for non-pharmacological measures, the evidence for drug-based and especially causal therapies remains very small. I have reworded the sentence appropriately.

Page 2, lines 53-54

The evidence for pharmacological and causal therapy in post-COVID-19 syndromes is still limited. .

  1. Line 48-51: Likewise, as an extension of above, there are multiple pharmacological therapies undergoing validation and in trials, so I would rephrase to reflect that. It is not the same as two years ago. This revised message would still flow into the following sentence (52-53).

Thank you for this important specification. I have now listed a few important trails.

Page 2, lines 54-57

However, several drug trials for the treatment of post-COVID-19 syndromes are ongoing, for example: Temelimab (NCT05497089), Montelukast (NCT04695704), Lithium (NCT05618587), nirmatrelvir plus ritonavir (NCT05576662), Low-Dose Naltrexone (NCT05430152), fampridine (NCT05274477).

  1. Line 58: there appears to be a rogue ‘2’ at the end of the sentence, I suspect this is a reference that has gone wrong.

Thank you very much. That was a formatting error from the next title. I have removed the 2.

  1. Line 66: you can remove the ‘the’ preceding post-COVID-19 syndrome.

Thank you very much, I have remove it.

  1. Line 72: I am not sure what this sentence adds

I have remove this sentence:” However here in the cohort study a referral bias has to be considered for our outpatient consultation” .

  1. You describe fatigue a few times in this sentence, which is slightly repetitive to the section above.

Thank you for the comment. I have corrected the sentence accordingly.

Page 2, lines 84-85

In the survey, study of Davis et al. 85.1% of respondents reported cognitive dysfunction, including poor attention and executive functioning (as problem solving, decision-making). [8].

  1. Line 76: I would add ‘such’ to ‘as’ in parenthesis.

I have add it.

  1. Line 80-82: What do these assessment tests and what is the conclusion gained from them?

I have added the corresponding tested and deficient neuropsychological domains.

Page 2, lines 91-93

In the study by Ortelli et al., patients with post-COVID-19 syndrome showed deficits in the Montreal Cognitive Assessment (MoCA) and Frontal Assessment Battery (FAB) for global cognition and executive functions and also in computerized attentive tasks for evaluating vigilance and executive attention compared to healthy individuals (age- and sex-matched healthy volunteers without COVID-19 infection).

  1. Line 82-85: Why not MoCA and why SDMT and DemTec. You need to provide an explanation for these statements.

I have adjusted it accordingly.

Page 2-3, Line 93-100

Current test recommendations for short cognitive tests in clinical routine are SDMT (for procession speed) or DemTEC (according to the recommendation of the Swiss Society of Insurance Medicine). [17, 18] The MoCA is likely to be poor at identifying cognitive deficits in patients with post-COVID-19 syndrome. [19] However, it must be kept in mind that these tests are not validated for post-COVID-19 syndrome and should only be used as an aid alongside history and clinical assessment for the detection of cognitive deficits. Otherwise, detailed neuropsychological testing is recommended. [17]

New Reference:

  1. 18. Biagianti B, Di Liberto A, Nicolo Edoardo A, Lisi I, Nobilia L, de Ferrabonc GD, et al. Cognitive Assessment in SARS-CoV-2 Patients: A Systematic Review. Front Aging Neurosci. 2022;14:909661.
  2. Lynch S, Ferrando SJ, Dornbush R, Shahar S, Smiley A, Klepacz L. Screening for brain fog: Is the montreal cognitive assessment an effective screening tool for neurocognitive complaints post-COVID-19? Gen Hosp Psychiatry. 2022;78:80-6.

  1. Line 86-87: Could you rephrase this sentence please, it is a bit clumsy.

I have rephrased this sentence.

Page 3, lines 101-102

The patients' quality of life and ability to work is influenced by symptoms such as fatigue and cognitive deficits.

  1. Line 89: What was the reason for inability to work?

The reason for the inability to work is the different symptoms of post-COVID-19 syndrome. The studies did not specify which symptom was the cause.

Page 3, lines 102-106

Previous studies of non-hospitalized patients with COVID-19 reporting that about 12-23% remain absent from work at 3 and 7 months after acute COVID-19 due to persistent symptoms. [8, 9] The inability to work lasts usually longer than 13 weeks. [5, 6] In our survey study more than half of the respondents (62.7%) with post-COVID-19 syndrome reported an inability to work, which lasted on average 26.6 weeks. [11]

  1. Could you describe some potential causes for the cognitive symptoms? Is it neural damage, specific impact of SARS-CoV-2 in the brain, the impact of fatigue, etc?

Thank you for the comment. Several factors play a role in the pathophysiology of post-COVID-19 syndrome. I have added a short paragraph in the introduction.

Page 1, lines 35-42

Pathogenesis of the post-COVID-19 syndrome is still unclear. Current evidence suggests several mechanisms that may be implicated in pathogenesis of post-COVID-19 syndrome: 1) Virus-related cellular changes associated with neurotropism. This mechanism could be responsible for olfactory dysfunction or autonomic nervous system dysfunction. 2) Dysregulated immune response in response to initial infection including autoimmune manifestations, activation of coagulation and fibrosis pathways or metabolic disturbances. An increase in multiple inflammatory mediators has also been demonstrated in patients with post-COVID-19 syndrome. 3) Occult viral persistence. [2]

New reference:

  1. Castanares-Zapatero D, Chalon P, Kohn L, Dauvrin M, Detollenaere J, Maertens de Noordhout C, Primus-de Jong C, Cleemput I, Van den Heede K. Pathophysiology and mechanism of long COVID: a comprehensive review. Ann Med. 2022 Dec;54(1):1473-1487.

  1. Line 93-96: some commas would help here to break up the sentence.

See point 18.

  1. Line 95-100: These three sentences could be combined into one

Thank you for your commentary. I have abbreviated and summarized the sentences.

Page 3, lines 109-110

Other frequently reported post-COVID-19 symptoms are headache, pain such as arthralgia, muscle pain and neuropathic pain and sleep disturbances. [10,11,20,21]

  1. Line 99: You have previously used the term SARS-CoV-2, so I recommend using SARS-CoV-1 for consistency.

Thank for your correction.

Page 3, lines 110-11

Neuropathic pain was also previously reported in patients recovered from SARS-CoV-1 a virus belonging to the family of coronaviruses, which caused to an outbreak in 2003. [21]

  1. Line 108-109: I am not sure what this adds – 12-40% is within the range of 9-50% previously mentioned.

Thank you for pointing this out, I have adjusted it accordingly.

Page 3, lines 119-120

Depending on the literature, the frequency of tachycardia syndromes, including POTS, varies between 9 and 50%.[10-11,22-24]. 

  1. Line 109-112: Which authors do you refer to? You? Also, you’ve already defined POTS so can use it. Again, overall, this is a long sentence that could be reduced to ‘As dysautonomia can exacerbate symptoms, including fatigue, it should be assessed for’. But you need to mention how. What is the best form of assessment?

Please excuse the unclear wording. I have now worded this sentence better.

Page 3, lines 120-122

Assessment, e.g. with Schellong test of these autonomic symptoms is particularly important, as symptoms such as postural tachycardia syndrome can be very limiting and can exacerbate fatigue.

  1. Line 115: I don’t like the first sentence of the paragraph and suggest you remove it.

I have removed the sentence.

  1. Line 115-123: Infact, I don’t like the entire paragraph. What are you trying to say? There isn’t a key message in this paragraph.

Thank you very much for the feedback. I have adjusted the paragraph accordingly.         

Page 3, lines 126-128

Currently, therapeutic options for post-COVID-19 syndrome are limited to non-drug measures or symptomatic therapy of symptoms. One of the greatest therapeutic challenges are the neurological symptoms.

  1. Line 127-139: This should be combined and act as your opening paragraph

See above

  1. Line 127: ‘the authors as board certified neurologists’ is an irrelevance.

See above

  1. Line 132: What are the implications regarding outpatient or inpatient settings? These are fundamentally different models with different levels of intensity, hours of rehabilitation, members of the multi-disciplinary team, resources…could you provide some discussion on one vs other?

Thank you for this important comment about rehabilitation. I have added a few important points.

Pages 3-4, lines 142-147

Initially, outpatient options should be prioritized. Inpatient rehabilitation should be evaluated if there is no improvement in symptoms following outpatient therapy. Inpatient rehabilitation can offer a more intensive and multimodal approach to therapy. Due to the variability of post-COVID-19 syndrome, a multimodal therapy approach such as inpatient rehabilitation can be advantageous.

  1. Line 133: I disagree regarding rarity. There has been a systematic review and meta analysis on impact of rehabilitation on fatigue (de Sire, A.; Moggio, L.; Marotta, N.; Agostini, F.; Tasselli, A.; Drago Ferrante, V.; Curci, C.; Calafiore, D.; Ferraro, F.; Bernetti, A.; Ozyemisci Taskiran, O.; Ammendolia, A. Impact of Rehabilitation on Fatigue in Post-COVID-19 Patients: A Systematic Review and Meta-Analysis. Appl. Sci. 2022, 12, 8593. https://doi.org/10.3390/app12178593) and one performed on fatigue and cognitive impairment (Felicia Ceban, Susan Ling, Leanna M.W. Lui, Yena Lee, Hartej Gill, Kayla M. Teopiz, Nelson B. Rodrigues, Mehala Subramaniapillai, Joshua D. Di Vincenzo, Bing Cao, Kangguang Lin, Rodrigo B. Mansur, Roger C. Ho, Joshua D. Rosenblat, Kamilla W. Miskowiak, Maj Vinberg, Vladimir Maletic, Roger S. McIntyre, Fatigue and cognitive impairment in Post-COVID-19 Syndrome: A systematic review and meta-analysis, Brain, Behavior, and Immunity,Volume 101, 2022, Pages 93-135, ISSN 0889-1591, https://doi.org/10.1016/j.bbi.2021.12.020.

Thank you for this important advice. I have adapted the paragraph accordingly.

Page 3, lines 131-133

In the last 2 years, there has been increased evidence on non-drug interventions for fatigue. [24, 25] In addition, lessons could be learned from the experience of treating fatigue in other conditions such as multiple sclerosis, chronic fatigue syndrome and cancer-associated fatigue

New References:

  1. De Sire, A., et al. Impact of Rehabilitation on Fatigue in Post-COVID-19 Patients: A Systematic Review and Meta-Analysis. Appl. Sci. 2022, 12, 8593.
  2. Ceban F et al. Fatigue and cognitive impairment in Post-COVID-19 Syndrome: A systematic review and meta-analysis, Brain, Behavior, and Immunity,Volume 101, 2022, Pages 93-135

  1. Line 141: In, not Im

Corrected

  1. Line 145-149: Check your tense

I have rephrased the sentence.

Page 4, lines 153-157

This therapy is targeted at helping patients to better manage the symptoms and their negative impact on activities of daily living (fatigue, pain, cognitive impairment, etc.). Furthermore, the patients should use the 3Ps rules in everyday life: Planning, Prioritization and Pause-management.

  1. Line 150: I would specify that IEME is used successfully in MS-fatigue and could be used for post-COVID-19 fatigue. The way this is phrased suggests it has been designed for post-COVID-19 fatigue.

Thank you for the comment. I have clarified the concept of energy management education (EME)

Page 4, lines 160-163

It should be noted that EME has been successfully used in MS fatigue, but has not yet been validated in post-COVID-19 syndrome. However, it should be used in the clinical routine of post-COVID-19 patients.

  1. Line 157: Earlier you said there was no evidence based treatment for post-COVID-19, but here you cite an RCT

The results are not yet published and therefore formally there is no evidence.

  1. Line 161-162: You can delete this sentence

I think this information is important; in everyday clinical practice, we are always confronted with study requests from patients. It is therefore important for non-scientifically active doctors to know where they can obtain information.

  1. This is a highly debated area. The most recent NICE guidance for ME/CFS have removed GET. There are multiple systematic reviews with different outcomes. I recommend you look into this area further. Furthermore, you need to conduct a baseline assessment prior to commencement, which you haven’t noted.

Thank you very much. Yes indeed, NICE has advised against graduated exercise therapy (GET). However, this decision was very sharply criticized in scientific communities. I have amended the paragraph.

Page 4, lines 179-186

Since the update of the guideline in 2020, GETis no longer recommended by NICE. NICE has not recommended GET because it considers this treatment to be ineffective and harmful, based on evidence from patient surveys and qualitative studies. [37] This decision has been widely criticized by the scientific community. [38, 39] The choice of ana-lysed studies was severely criticized. In addition, the omission of a recently published meta-analysis of the safety results in the ten published studies on GET was strongly objected to. [38]  This study showed that GET is safe as long as it is prescribed properly. [40]

New reference:

  1. National Institute for Health and Care Excellence. Guideline: myalgic encephalomyelitis (or encephalopathy)/chronic fatigue syndrome: diagnosis and management. Draft for consultation, November 2020. www.nice.org.uk/guidance/indevelopment/gid-ng10091/documents.
  2. Flottorp SA, Brurberg KG, Fink P, Knoop H, Wyller VBB. New NICE guideline on chronic fatigue syndrome: more ideology than science? Lancet. 2022;399(10325):611-3.
  3. Torjesen I. NICE backtracks on graded exercise therapy and CBT in draft revision to CFS guidance. BMJ. 2020;371:m4356.
  4. White PD, Etherington J. Adverse outcomes in trials of graded exercise therapy for adult patients with chronic fatigue syndrome. J Psychosom Res. 2021 Aug;147:110533. doi: 10.1016/j.jpsychores.2021.110533. Epub 2021 May 28. PMID: 34091377.

  1. Line 178: I don’t think that pacing involves increasing activity levels.

By pacing, ultimately you can improve the condition and thus increase the performance. I have adapted the sentence

Pacing is a prudent use of resources on a physical, mental and emotional level. The goal of pacing is to find the optimal and individual balance between rest periods and activation periods (physical, cognitive, emotional). [41, 42]

  1. Line 188: This sentence should sit with 179-181

I have shifted the sentence.

  1. You have provided no detail as how to perform heart rate monitoring, nor how should it be used in combination with pacing or GET. This section needs to be developed to provide a practical discussion of how HR monitoring should be used in post-COVID-19 syndrome, not just stating that it has been used in ME/CFS. Where does heart rate variability fit in?

Thank you for the comment. I have completed the paragraph.

Page 5, lines 207-211

Heart rate monitoring is a rehabilitation strategy that can be used by people to self-manage symptoms when living with post-COVID-19 syndrome. Heart rate monitoring is useful for people with fatigue to avoid PEM. It refers to the continuous measurement of heart rate with a heart rate monitor, is used to more accurately pace daily activities, and monitor the body’s responses to exertion.

Page 5, lines 222-224

Before using heart rate monitoring, autonomic dysfunction (e.g. postural tachycardia syndrome) in the context of post-COVID-19 syndrome must be ruled out, as this makes the use of these methods impractical.

  1. Should pharmacological therapy sit alongside non-pharmacological therapy, including psychological support, exercise intervention, patient education, and an understanding how to live well with persistent or chronic pain

Thank you for the specification. I have added it

Page 5, lines 231-232

In addition, non-medicinal measures including psychological support, exercise intervention, patient education, play an important role.

  1. What red flags? What medication regime/duration? What other elements need to be considered (such as PPI in chronic NSAID usage?) What do patients do on days when they can’t take medication? What other options are there?

I have added a review paper on red flags for headaches. Treatment of headaches should follow recommendations from national headache societies.

Page 5, lines 236-239

If there is no red flag [51], symptomatic treatment of the acute headache attacks are suggested (Non-steroidal anti-inflammatory drugs (NSAID), paracetamol or triptans in post-COVID-19 headaches that meet migraine criteria; according to recommendations of national headache societies.). [52, 53]

New reference:

  1. Do TP, et al. Red and orange flags for secondary headaches in clinical practice: SNNOOP10 list. Neurology. 2019 Jan 15;92(3):134-144.

  1. Who or what is Jacobsen? Can you expand this area and provide some citations please?

I have expand the paragraph.

Page 6, lines242-248

Also in the treatment of headaches, non-pharmacological measures such as progressive muscle relaxation, biofeedback and patients education play an important role. Progressive muscle relaxation by Jacobsen in particular has shown positive effects on headaches, e.g. migraines. During this technique, a total of 16 muscle groups of the body are identified and slightly tensed for a few seconds, followed by a 30 to 40 second relaxation phase. The practitioner concentrates on the perceived differences between the tension and the relaxation. In this way, the entire body is successively relaxed. [55]

New reference:

  1. Meyer B, et al. Progressive Muskelrelaxation nach Jacobson bei der Migräneprophylaxe : Klinische Effektivität und Wirkmechanismen [Progressive muscle relaxation according to Jacobson for migraine prophylaxis : Clinical effectiveness and mode of action]. Schmerz. 2018 Aug;32(4):250-258. German.

  1. Please don’t start the paragraph on sleep with medication. That is the last option that should be considered. Start with the first steps discussing sleep hygiene, then CBT, etc, then medication. Please specify which medication, how they work, and what regime

With all due respect, I think he misunderstood the sentence: “Although pharmacological therapy is commonly prescribed for insomnia, current guidelines do not recommend it to be first line therapy, especially due to adverse effects and lack of lasting effects. [56]”

References

 All COVID-19 related references should be within the last 12 months given the volume of ongoing research and updated findings that have been published.

The references were adjusted. We have made every effort to present the most up-to-date recommendations and results. However, data from 2022 is not available for all areas

Reviewer 2 Report

This article is a review that summarizes the most common neurological symptoms of post-COVID-19 syndrome and summarizes current knowledge about treatment options for neurological symptoms of post-COVID-19 syndrome, focusing on fatigue, pain, and sleep disorders. This manuscript provides a very well-updated review of post-COVID-19 syndrome and their therapeutics. In my opinion, it is suitable for publication in CTN.

Minor points:

1.         Line 82: can you clarify what the definition of "healthy individuals" was in the study by Ortelli, P., et al, J Neurol Sci, 2021? In 2021, it was very difficult to find a control group of individuals who had never developed COVID-19...

2.         Lines 89-92: could you please clarify whether these statistics were adjusted for pre-existing cognitive impairments in COVID-19?

Author Response

Response to reviewer 2

Dear Reviewer

Thank you for your valuable comments, which we appreciate. We hope we could address all your concerns properly.

  1. Line 82: can you clarify what the definition of "healthy individuals" was in the study by Ortelli, P., et al, J Neurol Sci, 2021? In 2021, it was very difficult to find a control group of individuals who had never developed COVID-19...

Thank you for the comment. The colleagues from Italy took twelve age- and sex-matched healthy volunteers without COVID-19 infection as a control population. Accordingly, I have made it more precise in the sentence.

Page 2, lines 89-93

In the study by Ortelli et al., patients with post-COVID-19 syndrome showed deficits in the Montreal Cognitive Assessment (MoCA), Frontal Assessment Battery (FAB) and also in computerized attentive tasks compared to healthy individuals (age- and sex-matched healthy volunteers without COVID-19 ).[12]

  1. Lines 89-92: could you please clarify whether these statistics were adjusted for pre-existing cognitive impairments in COVID-19?

These paragraphs refer to the ability to work. The respondents were asked how many hours or % work compared to before the COVID-19 disease. No questions were asked about pre-existing cognitive deficits or limitations.

Round 2

Reviewer 1 Report

Dear authors,

Thank you for taking on board the majority of my previous comments. The manuscript is improved, in my humble opinion, but I still feel the manuscript needs to be improved further prior to acceptance.

There are a few points below regarding this.

I also believe, as previously stated, that it would benefit from review by a native English speaker. 

Good luck

Was a literature search performed as part of this work, if so, some details regarding this would be helpful.

Line 20-34 – I think this paragraph in the introduction remains too long, and with the use of ‘first-person’ tense throughout (and, infact, throughout the paper). I would change this to neutral phrasing. 

Line 73-74 – again, this single sentence has 5 references (at least two of which are yours). Can you review this please.

Line 94-95 – please can you use SDMT and DemTEC in full on first use

In my previous review, I commented that you appeared to list your previous work without building on it. This hasn’t changed. You need to discuss your previous work in the context of this paper – or at the very least, add some commentary to it. The following examples are what I refer to: Line 81-82 - In our own studies, fatigue was the most common symptom with a frequency of 90.5- 93.2%.[10, 11]  Line 105-107 - In our survey study more than half of the respondents (62.7%) with post-COVID-19 syndrome reported an inability to work, which lasted on average 26.6 weeks. [11] Line 113 114 - In our own Swiss survey study, sleep disturbances were reported by 50.3% of the participants.

Line 135-138 – needs to be re written

You mention the 3P’s twice, and they are different. Are they the same, in which case, please harmonise them, or are they different, in which case, I suggest you make them more different. (from here; Line 155-157 Furthermore, the patients should use the 3Ps rules in everyday life: Planning, Prioritization and Pause-management, and here, Line 203 – 205 Pacing is often used as part of a set of energy conservation strategies called the 'Principle of Three P's', which include prioritization, planning and pacing.[47] )

Line 226-232 – please start with the non-pharmacological measures before the pharmacological measures. By listing it in the current order, it gives the impression that medication is the most important measure and the rest are token measures.

Line 235-248 – likewise!

Supplementary material / contributions / funding / review board statement / informed consent / data availability / acknowledgements / conflict of interest are all blank. Is that deliberate? 

References – please can you make them consistent? In particular, number of authors listed and the use of italic and bold font. Furthermore, certain references are still not complete, such as 11, 15, 16, 27 (and others) or in the wrong format, such as 25. Please refer to the journal’s style and use it. 

Supplementary material was the word version of the PDF file. Was there anything else to review?

Author Response

Response to reviewer 1

Dear Reviewer

Thank you for your valuable comments, which we appreciate. We hope we could address all your concerns properly. An English-language correction was performed.

  1. Was a literature search performed as part of this work, if so, some details regarding this would be helpful.

The literature search was carried out on PubMed and Google Scholar databases to

identify type of papers on long/post-COVID-19 syndrom . The search was conducted combining the terms

long/post-COVID-19, fatigue, cognitive symptoms, sleep disorders, post exertional malaise, autonomic dysfunction, postural tachycardia syndrome, treatment, prognosis.

  1. Line 20-34 – I think this paragraph in the introduction remains too long, and with the use of ‘first-person’ tense throughout (and, infact, throughout the paper). I would change this to neutral phrasing. 

I have adjusted the paragraph accordingly.

Page 1, lines 23-29

In the beginning, the acute effects and consequences of this infection were mainly considered. Later, however, the extent of the long-term consequences also became apparent. Indeed, not every patient fully recovers from the 2019 coronavirus disease (COVID-19) within weeks or months, resulting in post-COVID-19 syndrome. The World Health Or-ganization (WHO) has established an official definition for post-COVID-19 syndrome, i.e., the persistence of symptoms that cannot be explained by another disease 3 months after infection [1].

  1. Line 73-74 – again, this single sentence has 5 references (at least two of which are yours). Can you review this please.

I have reduced the references.

Page 2, lines 69-70

Fatigue, as known from other post-infectious syndromes, directly affects quality of life and participation in the activities of daily living (ADL). [8,10] 

  1. Line 94-95 – please can you use SDMT and DemTEC in full on first use

I have introduced the abbreviations

Page 2, lines 88-89

Current test recommendations for short cognitive tests in clinical routines include the Symbol Digit Modalities Test (SDMT; for procession speed) or the Demenz Detection Test (DemTEC; according to the recommendation of the Swiss Society of Insurance Medicine) [16, 17].

  1. In my previous review, I commented that you appeared to list your previous work without building on it. This hasn’t changed. You need to discuss your previous work in the context of this paper – or at the very least, add some commentary to it. The following examples are what I refer to:
    1. Line 81-82 - In our own studies, fatigue was the most common symptom with a frequency of 90.5- 93.2%.[10, 11]  
    2. Line 105-107 - In our survey study more than half of the respondents (62.7%) with post-COVID-19 syndrome reported an inability to work, which lasted on average 26.6 weeks. [11]
    3. Line 113 114 - In our own Swiss survey study, sleep disturbances were reported by 50.3% of the participants.

I have adjusted the passage:

  1. I have eliminated the sentence, as redundant
  2. Page 3, lines 96-103

The patients' quality of life and ability to work is influenced by symptoms such as fa-tigue and cognitive deficits. The data on the incapacity to work due to post-COVID-19 syndrome also vary widely depending on the literature, and the percentage of people ex-periencing an incapacity to work ranges from 23.3 to 62% [8-9,19]. This difference is most likely based on the definition of incapacity (partial vs. total). In the study of Davis et al., 23.3% of patients were totally unable to work and 48% were partially unable to work, whereas in our study, the percentage of people partially unable to work was 62% [8,19]. In general, the inability to work usually lasts longer than 13 weeks [8-9,19].

  1. Page 3, lines 110-118

Concerning sleep disturbances, quite similar results are found in the literature, with a prevalence of sleep disturbances of more than 50% [19,22]. The mean Insomnia Severity Index (ISI) was 13.0 points, and more than 1/3 of the respondents were above the threshold for clinical insomnia (defined by the ISI as ≥15 points) [19,22]. Excessive daytime sleep-ness (defined by the Epworth Sleepiness Scale (ESS) as ≥11 points) was reported in 56.3% of the respondents [19]. Especially with such elevated ESS scores, obstructive sleep apnea syndrome (OSA) must be considered in the differential diagnostic workup for post-COVID-19 syndrome. In fact, a current study has shown a high rate of OSA in COVID-19-patients that exceeds the expected prevalence of OSA [23].

New references:

  1. El Sayed, S.; Gomaa, S.; Shokry, D.; Kabil, A.; Eissa, A. Sleep in post-COVID-19 recovery period and its impact on differ-ent domains of quality of life. Egypt J Neurol Psychiatr Neurosurg 2021, 57, 172, doi:10.1186/s41983-021-00429-7.
  2. Schwarzl, G.; Hayden, M.; Limbach, M.; Schultz, K. The prevalence of Obstructive Sleep Apnea (OSA) in patients recov-ering from COVID-19. ERJ Open Research 2021, 7, 24, doi:10.1183/23120541.sleepandbreathing-2021.24.
  3. Line 135-138 – needs to be re written

The sentence was reworded.

Page 3, lines 136-138

In contrast to the role of exercise in MS fatigue, in post-COVID-19 patients, due to post-exertional malaise, the anaerobic threshold does not need to be exceeded.

  1. You mention the 3P’s twice, and they are different. Are they the same, in which case, please harmonise them, or are they different, in which case, I suggest you make them more different. (from here; Line 155-157 Furthermore, the patients should use the 3Ps rules in everyday life: Planning, Prioritization and Pause-management, and here, Line 203 – 205 Pacing is often used as part of a set of energy conservation strategies called the 'Principle of Three P's', which include prioritization, planning and pacing.[47] )

Pacing and pause management are used as synonyms in this context. However, I have unified it.

Page 5, lines 205-206

Pacing is often used as part of a set of energy conservation strategies called the “Principle of Three P's”, which include prioritization, planning, and pause management[48]. 

  1. Line 226-232 – please start with the non-pharmacological measures before the pharmacological measures. By listing it in the current order, it gives the impression that medication is the most important measure and the rest are token measures.

I adjusted the paragraph and took the non-medication measure first.

Page 5, line 228-233

Non-medicinal measures including psychological support, exercise intervention, and patient education play an important role. As a general rule, if symptoms are frequent and other underlying causes—such as vitamin deficiencies, diabetes mellitus, iron deficiency or thyroid dysfunction—have been ruled out, neuropathies can be treated with pharma-cological treatments including, but not limited to, pregabalin, gabapentin, or duloxetine, as well as local capsaicin treatment [51, 52].

  1. Line 235-248 – likewise!

 I adjusted the paragraph and took the non-medication measure first.

Page 5-6, line 237-253

In the treatment of headaches, non-pharmacological measures such as progressive muscle relaxation, biofeedback, and patient education also play an important role. In par-ticular, progressive muscle relaxation as defined by Jacobsen has shown positive effects on headaches, e.g., migraines. During this technique, a total of 16 muscle groups of the body are identified and slightly tensed for a few seconds, which is followed by a 30 to 40 second relaxation phase. The practitioner concentrates on the perceived differences be-tween the tension and the relaxation. In this way, the entire body is successively relaxed [53].

If there are no red flags [54], the symptomatic treatment of the acute headache attacks are suggested (such as non-steroidal anti-inflammatory drugs (NSAIDs), paracetamol, or triptans in post-COVID-19 headaches that meet migraine criteria; this is according to recommendations of the International Headache Society) [55, 56]. However, it is very important to make the patient aware that painkillers should be taken less than 10-12 days per month in order to minimize the risk of an analgesic-induced headache [56].In the event of chronic debilitating headaches which do not improve with standard analgesics, a disease-modifying treatment specific to the headache type should be instituted [55,56].

  1. Supplementary material / contributions / funding / review board statement / informed consent / data availability / acknowledgements / conflict of interest are all blank. Is that deliberate? 

Supplement material is integrated in Word (Figure 1). The authors have no conflict of interest related to this publication.

  1. References – please can you make them consistent? In particular, number of authors listed and the use of italic and bold font. Furthermore, certain references are still not complete, such as 11, 15, 16, 27 (and others) or in the wrong format, such as 25. Please refer to the journal’s style and use it. 

References have been adjusted.

  1. Supplementary material was the word version of the PDF file. Was there anything else to review?

Supplement material is integrated in Word (Figure 1).
